# Probiotics-Fermented *Grifola frondosa* Total Active Components: Better Antioxidation and Microflora Regulation for Alleviating Alcoholic Liver Damage in Mice

**DOI:** 10.3390/ijms24021406

**Published:** 2023-01-11

**Authors:** Xiao-Yu He, Yu-Xian Zhu, Xiao-Qin Jiang, Fu-Rong Zhu, Yi-Juan Luo, Yu-Yang Qiu, Zi-Rui Huang, Bin Liu, Feng Zeng

**Affiliations:** 1National Engineering Research Center of JUNCAO Technology, Fuzhou 350002, China; 2College of Food Science, Fujian Agriculture and Forestry University, Fuzhou 350002, China; 3School of Agriculture and Biology, Shanghai Jiao Tong University, Shanghai 200240, China; 4Engineering Research Centre of Fujian Subtropical Fruit and Vegetable Processing, Fujian Agriculture and Forestry University, Fuzhou 350002, China

**Keywords:** alcoholic liver damage, *Grifola frondosa*, probiotic fermented, antioxidant, 16S rRNA gene sequencing, intestinal microbiota, *Lactobacillus*

## Abstract

Alcoholic liver damage is caused by long-term drinking, and it further develops into alcoholic liver diseases. In this study, we prepared a probiotic fermentation product of *Grifola frondosa* total active components (PFGF) by fermentation with *Lactobacillus acidophilus*, *Lactobacillus rhamnosus*, and *Pediococcus acidilactici*. After fermentation, the total sugar and protein content in the PFGF significantly decreased, while the lactic acid level and antioxidant activity of the PFGF increased. Afterward, we investigated the alleviating effect of PFGF on alcoholic liver injury in alcohol-fed mice. The results showed that the PFGF intervention reduced the necrosis of the liver cells, attenuated the inflammation of the liver and intestines, restored the liver function, increased the antioxidant factors of the liver, and maintained the cecum tissue barrier. Additionally, the results of the 16S rRNA sequencing analysis indicated that the PFGF intervention increased the relative abundance of beneficial bacteria, such as *Lactobacillus*, *Ruminococcaceae*, *Parabacteroids*, *Parasutterella*, and *Alistipes*, to attenuate intestinal inflammation. These results demonstrate that PFGF can potentially alleviate alcoholic liver damage by restoring the intestinal barrier and regulating the intestinal microflora.

## 1. Introduction

Alcoholic liver damage (ALD) is the damage to liver function, and it is caused by drinking. ALD further develops into alcoholic liver diseases such as alcoholic fatty liver, inflammatory hepatitis, hepatic cirrhosis, and hepatic fibrosis [1]. The Global Status Report on Alcohol and Health (2018) published by the World Health Organization indicates that 3 million deaths occur every year as a result of harmful alcohol use [2]. Oxidative stress is the major cause of ALD. Oxidative stress is caused by reactive oxygen species (ROS) from hepatic alcohol metabolizers. Overwhelming oxidative stress causes cellular damage and produces hepatotoxic substances such as malondialdehyde [3]. Oxidative stress can also reduce the content of antioxidant factors such as superoxide dismutase and glutathione [4]. Chronic alcohol consumption damages the intestinal barrier, which leads to intestinal bacterial dysbiosis and increases the systemic levels of bacterial products. Harmful substances such as lipopolysaccharide transfer from the intestines to the liver and cause liver inflammation [5]. Currently, stopping alcohol consumption is the most helpful method of alleviating ALD, as considerable side effects are associated with the use of chemical drugs [6]. Thus, the search for less toxic treatments is a priority.

More recently, some studies have demonstrated an association between the intestinal microbiota and ALD [7]. The effect of alcohol on the intestinal microbiota is called dysbiosis [8]. Alcohol can increase the abundance of endotoxin-producing microbiota and decrease the abundance of short-chain fatty acids (SCFAs)-producing microbiota and anti-inflammatory microbiota [9,10]. Negative alterations in the intestinal microbiota increase the liver damage through the gut–liver axis [11]. In general, probiotics refer to live microorganisms that, when they are administered in adequate amounts, confer health benefits to the host. Probiotics can boost the immunological system, protect the gut, and positively regulate the intestinal microbiota [12]. In addition, probiotic fermentation can improve the functional characteristics of food. Fermented dairy foods have been consumed for many years as traditional probiotic fermented foods, and they have proven benefits [13]. At present, many studies of nondairy probiotic fermented foods have also been conducted. Zhang et al. found that blueberry juice fermented by *Lactobacillus plantarum* had increased antioxidant activity [14]. Hu et al. found that fermented carrot juice could attenuate type 2 diabetes by mediating the intestinal microbiota in rats [15].

*Lactobacillus* has a long history use as fermentation-engineering bacteria because they are beneficial in the prevention of intestinal diseases and the regulation of intestinal flora [16]. The strains that are commonly studied worldwide include: *Lactobacillus acidophilus*, *Lactobacillus rhamnosus*, and *Pediococcus acidilactici*, etc. [17]. *Lactobacillus rhamnosus* is one of the most widely studied probiotic strains, and it can attenuate inflammation, improve hydrogen-peroxide-induced damage, reduce the viability of pathogenic bacteria, regulate the intestinal microbiota, and regain intestinal barrier function, etc. [18]. Forsyth et al. found that gavage with *Lactobacillus rhamnosus* GG reduced the severity of alcohol-induced gut hyperpermeability and alcohol-induced tissue and systemic oxidative stress to ameliorate alcohol-induced gut leakiness and liver injury [19]. *Lactobacillus acidophilus* also has antibacterial and anti-inflammatory activities [20]. Lv et al. found that *Lactobacillus acidophilus* could alleviate D-GalN-induced liver injury in rats through downregulating the expression of the infection- and inflammation-related pathways [21]. *Pediococcus acidilactici* is a safe probiotic that is widely used in the food industry [22]. Zhang et al. found that *Pediococcus acidilactici* FZU106 could significantly reduce the accumulation of excessive lipid in the liver caused by high-fat diet feeding in mice [23]. Bikheet et al. found that *Pediococcus acidilactici*-fermented milk could treat the liver fibrosis caused by CCl_4_ [24].

*Grifola frondosa* is recognized as a functional food, and it has been extensively studied for its antitumor, immunoregulation, anti-inflammation, and hepatoprotective effects [25]. *Grifola frondosa* has ameliorating effects on liver damage. Chen et al. found that *Grifola frondosa* polysaccharides could inhibit liver lipid peroxidation in rats [26]. In our previous studies, we have found that 95% and 55% *Grifola frondosa* ethanol extract and *Grifola frondosa* heteropolysaccharide could improve glucose and lipid metabolism disorders by attenuating tissue inflammation, reducing the occurrence of liver lesions, and regulating the intestinal microbiota [27,28,29]. Therefore, *Grifola frondosa* has anti-inflammation and hepatoprotective abilities and improves the structure of the intestinal microbiota, so it shows potential for improving ALD. However, a few studies have been conducted on the ameliorating effects of *Grifola frondosa* on ALD. In this study, we describe the total active components of a probiotic fermentation of *Grifola frondosa,* and we determined the ability of the total active components of *Grifola frondosa* and its fermentation products to improve ALD.

## 2. Results and Discussion

### 2.1. Composition Analysis and Antioxidant Activities In Vitro

Data about the total sugar content (TS), reducing sugar content (RS), protein content (PR), total flavonoid content (TF), total polyphenol content (TP), and lactic acid content (LC) are shown in Table 1. Compared with the composition of the unfermented *Grifola frondosa* total active components (GF) sample, the TS, RS, and PR in the fermentation products of *Grifola frondosa* total active components (PFGF) group were significantly lower, TF and TP were not significantly different, while the LC was significantly higher. The results of the 1,1-diphenyl-2-picrylhydrazyl (DPPH) free radical scavenging activity assay, the Fe^2+^ chelating activity assay (FCA), and the reducing power assay (RPA) are also shown in Table 1. The PFGF showed significantly higher antioxidant activities than GF did.

*Grifola frondosa* is rich in polysaccharides and proteins. In our previous studies, we found that *Grifola frondosa* extracts are also rich in flavonoids and polyphenols. These compositions are the fermentation substrates for probiotics, and they are the source of the antioxidant activity of *Grifola frondosa* [27]. *Lactobacillus* and *Pediococcus acidilactici* can use carbohydrates as an energy source to produce organic acids such as lactic acid [30,31]. During GF fermentation, the probiotics used the sugars and proteins in the fermentation system and produced a large amount of lactic acid.

Lactic acid is one of the important metabolites of *Lactobacillus.* Lactic acid can inhibit the growth of pathogenic bacteria by reducing the intestinal pH [32]. Wang et al. found that lactic acid has an antibacterial ability and can completely inhibit the growth of *Salmonella enteritidis*, *E. coli,* and *L. monocytogenes* cells [33]. This proved that lactic acid can maintain the homeostasis of intestinal flora by inhibiting the growth of harmful bacteria, thereby attenuating intestinal inflammation.

The antioxidant activity of GF also increased after fermentation, which is consistent with previous findings [14,34]. *Lactobacillus* can metabolize reducing substances, such as glutathione (GSH), which can play an antioxidant role [35]. In addition, the preservation of flavonoids and polyphenols and the production of bioactive polysaccharides and peptides with antioxidant activity that were produced by the enzymatic hydrolysis of probiotics also increased the antioxidant capacity of PFGF [36].

### 2.2. Effects of PFGF on Body Weight and Liver Index

The results of the body weight, liver weight, and liver index are shown in Figure 1A–D. After the alcohol gavage, the weight of the mice significantly decreased. After 6 weeks of intragastrical administration, compared with the model (EtOH) group, the probiotic (P) and GF groups showed no significant recovery in their body weight, whereas that of the PFGF group significantly increased. Compared with the EtOH group, the mice in the P, GF, and PFGF groups showed significantly decreased liver weights and liver indexes.

Body weight is an important parameter indicating the physical state of the mice, and it can reflect the intestinal absorption of nutrients [2,37]. The liver index reflects the degree of liver damage [38]. The recovery of body weight indicated that PFGF could better restore the function of intestinal absorption, which might be related to the direct intake of lactic acid with PFGF [39]. The decline in the liver index indicated that P, GF and PFGF could decrease the amount of liver damage.

### 2.3. Effects of PFGF on Biochemical Parameters of Serum and Liver

The data about the serum total cholesterol (TC), triglyceride (TG), endotoxin (LPS), aspartate aminotransferase (AST), alanine aminotransferase (ALT), liver superoxide dismutase (SOD), GSH, LPS, and malondialdehyde (MDA) are shown in Figure 2A–I. After 6 weeks of intragastrical administration, compared with the M group, all of the biochemical parameters of the P, GF, and PFGF groups improved. Compared with the GF group, the TC of the PFGF group significantly decreased. Compared with that of the P group, the SOD of the PFGF group was also significantly increased.

The TC and TG in the serum reflect the liver’s function. The AST and ALT in the serum reflect the levels of hepatocyte necrosis [40]. LPS and MDA are major factors in inflammation and liver damage, whereas SOD and GSH are antioxidative factors that protect the liver against the excessively increased levels of LPS and MDA [41,42].

The amelioration of the biochemical parameters in this study indicated that P, GF, and PFGF had a recovery effect on the liver damage, of which PFGF produced the best effect. The metabolites of P in the intestinal tract, GF and PFGF, show a favorable antioxidant activity, which can reduce the alcohol-induced intestinal inflammation [43]. Both the probiotics and *Grifola frondosa* have a positive intestinal microbiota regulation ability [23,44,45]. They have the ability to attenuate the intestinal inflammation caused by the alcohol-induced flora disorder. The amelioration of intestinal inflammation decreases intestinal permeability, preventing harmful bacterial products from entering the blood system by crossing the intestinal barrier, recovering the liver from injury, and further improving the liver’s function.

### 2.4. Effects of PFGF on Liver and Cecal Tissue

The histopathology results of the liver and cecal tissue are shown in Figure 3. The clear staining of the liver cells of the normal control (NC) group showed that they were orderly. There were many binucleate cells, but there were no necrotic cells. The EtOH group showed obvious inflammatory infiltration in the liver tissue. The cells showed notable ulceration and necrosis. We observed almost no binuclear cells. With intraperitoneal administration, we observed no notable ulceration or necrosis of the liver cells in the P, GF and PFGF groups; the liver tissue remained in good condition.

The cecal tissue of the NC group was clearly stained. The cecal villi were intact and neatly arranged, and the cecal wall was intact, with a thin lipid layer. The cecal tissue of the EtOH group showed substantial inflammation. The cecal villi were broken, the cecal wall was damaged, and the lipid layer was thickened. With intraperitoneal administration, the integrity and orderliness of the intestinal villi in the P, GF and PFGF groups were maintained, as was the integrity of the intestinal wall. A thin lipid layer was maintained in both the P and PFGF groups, but the GF group showed a thickening of the lipid layer.

The histopathology of the cecal tissue can reflect the states of the intestinal barrier and inflammation in the intestine. In our previous study, the compounds added with GF improved the jejunum tissue and reduced the level of IL-6 in type 2 diabetes mice, which indicates that GF can alleviate intestinal inflammation [45]. The results of the cecal tissue sections showed that the P, GF, and PFGF treatments had protective effects on the intestinal barrier. These protective effects were also reflected in the liver tissue sections. However, the cecal tissue in the GF group also showed inflammation, which suggested that P and PFGF could better protect the intestinal barrier.

### 2.5. Effects of PFGF on Composition of Intestinal Microbiota

The results of the 16S rRNA gene sequencing and the values of the relative abundance of *Firmicutes* and *Bacteroides* are shown in Figure 4A–C. The relative abundance of intestinal microbiota at phylum and genus are shown in Appendix A. At the phylum level (Figure 4A), the intestinal microbiota was mainly composed of *Firmicutes*, *Bacteroidetes*, *Proteobacteria*, and *Actinbacteria*. Compared with those of the M group, the relative abundance of *Firmicutes* in the GF and PFGF groups was significantly lower, but the relative abundance of *Bacteroides* in the NC, P, GF, and PFGF group were significantly higher. At the genus level (Figure 4B), the intestinal microbiota was mainly composed of *Faecalibaculum*, *Lactobacillus*, *Dubosiella*, *Bacteroides*, and *Alloprevotella*. The relative abundance of *Lactobacillus* in the P and PFGF groups was significantly higher than that in the other groups. Figure 4C shows that the *Firmicutes* and *Bacteroides* value of the EtOH group was significantly higher than that of the NC group. After the gavage treatment, the values for the P, GF, and PFGF group significantly decreased.

*Lactobacillus acidophilus*, *Lactobacillus rhamnosus*, and *Pediococcus acidilactici* all belong to *Lactobacillus* and *Firmicutes*. The increase in the relative abundance of *Lactobacillus* in the P and PFGF groups indicated that the probiotics had successfully colonized the mouse intestines. The intestinal microbiota can reflect the health status of the intestine; the *Firmicutes*/*Bacteroides* value can especially reflect intestinal inflammation and the intestinal permeability [46].

The PCA score plot and hierarchical clustering plot are shown in Figure 4D,F, which shows a substantial separation between the NC, EtOH, PFGF, P, and GF groups. The NC group is clustered on the negative half of the axis of the first principal component (PC1), while the EtOH group is clustered on the negative half of the axis of PC2, the P and GF groups are clustered on the positive half of the axis of PC1, and the PFGF group is clustered on the positive half of the axis of PC2. The hierarchical clustering plot shows that the samples of each group were clustered in the same class, and they were significantly different from the other groups.

PCA and hierarchical clustering are often used to analyze the similarities and differences between groups of intestinal microbiotas [47]. In the PCA score plot, compared with the NC group, the EtOH group significantly shifted to the negative axis of PC2, indicating that alcohol intake affected the host’s intestinal microbiota. The P, GF, and PFGF groups were distant from the NC and EtOH groups, indicating that P, GF, and PFGF all had the ability to regulate the intestinal microbiota. Similar results are also reflected in the hierarchical clustering plot.

### 2.6. Correlations of Intestinal Microbiota with Biochemical Parameters

The taxa with an LDA score threshold of >3.0 are shown in Figure 5A as a bar chart. *Alistipes*, *Odoribacter*, and *Angelakisella*, etc., were the characteristic microbes in the NC group. *Bacteroides*, *Desulfovibrio*, and *Blautia*, etc., were the characteristic microbes in the EtOH group. *Lactobacillus* was the characteristic microbe in the P group. *Faecalibaculum*, *Alloprevotella*, and *Bifidobacterium* were the characteristic microbes in the GF group. *Unidentified_Ruminococcaceae* and *Parabacteroides*, etc., were the characteristic microbes in the PFGF group.

A heatmap of the correlations between the significantly different intestinal microbiota and biochemical parameters and a visualization of the correlation network are shown in Figure 5B,C. The correlation heatmap shows that *Lactobacillus*, *unidentified_Ruminococcaceae*, *Parabacteroides*, *Odoribacter*, *Alistipes*, and *Angelakisella* were positively correlated with the serum TC, TG, AST, ALT, and LPS and liver MDA, LPS, and index, and they were negatively correlated with the liver GSH and SOD. The opposite results were reflected in *Bacteroides*, *Desulfovibrio*, and *Blautia*. Squares indicating significant correlations (|r| > 0.5 and *ρ* < 0.01) are marked with an asterisk, and their correlations are shown in the visualization of the correlation network. These intestinal microbiotas that were significantly associated with biochemical indicators were defined as key intestinal microbiota, and their relative abundances are shown in Appendix A.

In Section 1, we mentioned many benefits of *Lactobacillus*. Except for the opposite correlation between the relative abundance of *Lactobacillus* and liver LPS, we found no significant correlation between the relative abundance of *Lactobacillus* and the biochemical indicators in this study. This may have been caused by the relative abundance of *Lactobacillus* in the EtOH group which was also significantly increased compared with that in the NC group. Bull et al. also found that chronic alcohol feeding resulted in an increase in the relative abundance of *Lactobacillus* [48]. Recently, a study showed that not all *Lactobacillus* have positive effects [49]. Chronic alcohol feeding may increase the relative abundance of harmful *Lactobacillus*.

*Ruminococcaceae* is one of the hallmark bacteria of intestinal inflammation, which is reduced in colitis patients [50]. In this study, alcohol intake significantly reduced the relative abundance of *Ruminococcaceae*, but the PFGF treatment restored its relative abundance to normal levels. Shang et al. [51] found that the abundance of *Ruminococcaceae* in the intestinal microbiota negatively correlated with liver cirrhosis, nonalcoholic fatty liver, and increased intestinal permeability. We obtained similar results in our study. Lee et al. [52] also reported that colonization of *Ruminococcus faecis* and *Ruminococcus bromii* significantly improved the liver injury in nonalcoholic fatty liver mice.

*Parabacteroides*, as one of the core flora of human intestinal flora, has the ability to participate in carbohydrate metabolism and secrete short-chain fatty acids [53]. *Parabacteroides* was the characteristic microbe of the PFGF group, but in this study, we found a negative correlation between the relative abundance of *Parabacteroides* and the parameters of the liver’s function. This may have occurred because the relative abundance of *Parabacteroides* in the EtOH group was significantly higher than that in the NC, P, and GF groups. Li et al. and Liu et al. also found that the relative abundance of *Parabacteroides* in the intestinal flora of mice increased after alcohol feeding [54,55]. Their findings all showed that an increase in the relative abundance of *Parabacteroides* could improve the biochemical parameters of chronically alcohol-exposed mice.

*Parasutterella* is a core component of the human and mouse intestinal microbiota. Ju et al. found that *Parasutterella* can produce succinate, which can recover colitis [56,57]. Huang et al. found that *Parasutterella* plays a positive role in the production of short-chain fatty acids and the reduction in inflammation [49]. In this study, P, GF, and PFGF could significantly increase the relative abundance of *Parasutterella*, and we found a strong relationship between the abundance of *Parasutterella* and the biochemical parameters. Mu et al. found that the relative abundance of *Parasutterella* negatively correlated with TC and TG, which is consistent with our findings [58].

*Alistipes* and *Odoribacter* were common beneficial intestinal bacteria in our previous studies. Wang et al. and Pan et al. found that the relative abundance of *Alistipes* positively correlated with SOD and GSH, and it was negatively correlated with TC, TG, AST, and ALT [27,29]. *Odoribacter* is short-chain fatty-acids-producing bacteria, and it is thought to promote blood lipid metabolism [59,60]. *Desulfovibrio* is an intestinally harmful bacteria. They are a class of anaerobic bacteria that can reduce sulfate to producing H_2_S. Endogenic H_2_S can poison intestinal epithelial cells, causing intestinal leakage, abdominal pain, and chronic inflammation, etc. [61]. In this study, the PFGF restored the relative abundance of *Alistipes* and reduced the relative abundance of *Desulfovibrio.*

These results indicated that PFGF can regulate the microbial population structure, restore or increase the relative abundance of beneficial bacteria, and reduce the relative abundance of harmful bacteria, thus alleviating the effects of alcoholic liver damage.

## 3. Materials and Methods

### 3.1. Extraction of Active Components

We improved upon our previously reported method of extracting total active ingredients from *Grifola frondosa* (provided by the China National Engineering Research Center of JUNCAO Technology, Fujian Agriculture and Forestry University). In short, we extracted *Grifola frondosa* at different concentrations of alcohol under an ultrasound (45 kHz, 300 W) for one hour. We collected the supernatant, and we collected and dried the residue for the next extraction. After extraction with 95% (*v*/*v*) ethanol (10:1, *v*/*w*, 45 °C), 55% (*v*/*v*) ethanol (10:1, *v*/*w*, 45 °C), and deionized water (30:1, *v*/*w*, 75 °C), we collected the residue and added deionized water (10:1, *v*/*w*, 55 °C). We then adjusted the pH to 7.5 with NaHCO_3_, and we used papain (purchased from Solarbio Technology Co., Ltd., Beijing, China) to obtain the hydrolyzed protein. We collected the supernatant after 4 h of enzymatic hydrolysis. Finally, we filtered, concentrated, and freeze-dried all of the supernatants to obtain the *Grifola frondosa* total active components.

### 3.2. Probiotic Strain and Inoculum Preparation

*Lactobacillus rhamnosus* (CICC number 20255, which we purchased from China Center of Industrial Culture Collection, Beijing, China) and *Pediococcus acidilactici* (CICC number 10146, purchased from China Center of Industrial Culture Collection, China) were recovered and inoculated into an MRS medium (Merck & Company, Inc., Darmstadt, Germany). We recovered *Lactobacillus acidophilus* (Provided by College of Food Science, Fujian Agriculture and Forestry University, Fuzhou, China), which we inoculated into Bengal red medium (Merck & Company, Inc., Darmstadt, Germany), and cultured the sample at 37 °C for 24 h, and then centrifuged it at 800× *g* for 10 min at 4 °C. We collected and resuspended the precipitate with normal saline. Finally, we mixed all of the probiotics at 1:1:1 by mass and resuspended them. We calculated the bacterial concentration by the plate-counting method, and we diluted the probiotics with normal saline to 1 × 10^9^ CFU/mL for later use.

### 3.3. Preparation of Probiotic Fermentation Products

The fermentation system included 0.4 g of glucose, 99 mL of deionized water, and 10 g of pasteurized *Grifola frondosa* total active components. We adjusted the pH of the fermentation system to 6.5 with NaHCO_3_. Then, we added 1 mL of *Lactobacillus rhamnosus*, *Pediococcus acidilactici*, and *Lactobacillus acidophilus* bacterial solutions. After fermentation at 37 °C for 48 h and after we reached a total bacterial concentration > 1 × 10^9^ CFU/mL, we collected the suspension to obtain the fermentation products of *Grifola frondosa* total active components (PFGF). The unfermented *Grifola frondosa* total active components (GF) served as the control.

### 3.4. Composition Determination

We measured the total sugar content (TS), reducing sugar content (RS), protein content (PR), total flavonoid content (TF), total polyphenol content (TP), and lactic acid content (LC) in GF and PFGF. We used the phenol–sulfuric acid method to determine the TS [62]. We used the dinitrosalicylic acid method was used to determine the RS [63]. We determined the PR according to a BCA kit (Lablead Biotechnology Co., Ltd., Beijing, China). We determined the TF and TP by referring to the method of Toro-Uribe et al. [64]. We determined the lactic acid content according to an LD kit (Nanjing Jiancheng Bioengineering Co., Ltd., Nanjing, China). We purchased the reagents used from Sinopharm Chemical Reagent Co., Ltd. (Beijing, China).

### 3.5. Determination of Antioxidant Activities In Vitro

For the 1,1-diphenyl-2-picrylhydrazyl (DPPH) free radical scavenging activity assay and the Fe^2+^ chelating activity assay (FCA) of GF and PFGF, we referred to the method of Nilgün Ozdemir et al. [65]. For the reducing power assay (RPA), we referred to the method of Cao et al. [66]. We purchased the reagents used from Sinopharm Chemical Reagent Co., Ltd. (Beijing, China).

### 3.6. Animal Model, Administration, and Sample Collection

We purchased 50 male ICR mice from Wu’s Animal Center (Fuzhou, China), which we housed in a chamber (23 ± 2 °C) with a 12 h dark/light cycle and free access to food and water. After 1 week of acclimatization, we randomly divided the mice into 5 groups. We intragastrically administered all of the mice in the morning, and we administered the probiotic (P, *n* = 10) group, the unfermented *Grifola frondosa* total active components (GF, *n* = 10) group, and the fermentation products of *Grifola frondosa* total active components (PFGF, *n* = 10) group by gavage with probiotics (100 mg/kg), GF (100 mL/kg) and PFGF (100 mL/kg), respectively. We administered the normal control (NC, n = 10) group and model (EtOH, n = 10) group by gavage with equal volumes of normal saline instead. We administered the second gavage to all of the mice in the afternoon, and we administered the EtOH, P, GF, and PFGF groups by gavage with 52% liquor alcohol (10 mL/kg, 36% liquor alcohol in the first week). The NC group was administered by gavage with an equal volume of normal saline instead. We measured their body weight every 2 weeks. After 6 weeks of gavage, we sacrificed all of the mice by cervical dislocation after weighing, and we retained the serum, liver, cecum, and intestinal contents, which we stored at −80 °C until further use.

### 3.7. Biochemical Parameters of Serum and Liver

We weighed the mouse livers, and we calculated liver index (the ratio of liver weight to body weight). We determined the contents of superoxide dismutase (SOD), glutathione (GSH), endotoxin (LPS), and malondialdehyde (MDA) using a lipopolysaccharide ELISA kit, a superoxide dismutase assay kit, a malondialdehyde assay kit, and a reduced glutathione assay kit, respectively. We centrifuged the serum and removed the supernatant. We determined the contents of total cholesterol (TC), triglyceride (TG), LSP, aspartate aminotransferase (AST), and alanine aminotransferase (ALT) using an aspartate aminotransferase assay kit, an alanine aminotransferase assay kit, a lipopolysaccharide ELISA kit, a total cholesterol assay kit, and a triglyceride assay kit, respectively. We purchased all of the kits from Nanjing Jiancheng Bioengineering Co., Ltd. (Nanjing, China).

### 3.8. Preparation of Liver and Cecal Tissue Sections

We processed liver and cecal samples according to the method of Huang et al. [47], which we then stained with hematoxylin and eosin. We observed the liver and cecal tissue sections under an optical microscope (Nikon, Tokyo, Japan), and we performed the histopathological evaluation according to Roth et al. [67].

### 3.9. High-Throughput Sequencing Analysis of Intestinal Microbiota

We extracted genomic DNA from intestinal contents according to the method of Ge et al. [68]. We amplified the V3−V4 region of the bacterial 16S rRNA gene using universal primers 338F and 806R. We produced the 16S rRNA gene sequencing libraries of bacteria with a TruSeq DNA PCR-Free Sample Preparation Kit (Illumina, San Diego, CA, USA) [69]. Novogene Co., Ltd. (Beijing, China) provided the Illumina NovaSeq PE250 platform for us to complete high-throughput sequencing.

We drew the rank abundance curves using the “ggalt” and “BiodiversityR” packages in R software (ver. 4.1.0). We conducted a principal component analysis (PCA) of the intestinal microbiota at the genus level with SIMCA-14.1. We used the Huttenhower Lab Galaxy server (http://huttenhower.sph.harvard.edu/lefse/, accessed on 3 August 2022) to analyze variation in the different experimental groups with the linear discriminant analysis (LDA) effect size (LEfSe) algorithm (threshold > 3). We calculated the relationship between the relative abundance of the biochemical parameters and intestinal microbiota by Spearman’s rank correlation analysis, and we visualized the results using the “psych” and “pheatmap” packages in R software (ver. 4.1.0). We plotted the correlation network with Cytoscape software (ver. 3.7.1).

### 3.10. Statistical Analysis

We analyzed all of the data with SPSS statistics 17.0, and we drew figures using GraphPad Prism 9. All of the data are shown as the mean ± standard deviation (SD). We used Student’s t-test to determine statistical significance between two groups. We considered *p* < 0.05 statistically significant. We used one-way analysis of variance (ANOVA) with Tukey’s correction to determine the statistical significance between multiple groups. We considered *p* < 0.05 statistically significant.

## 4. Conclusions

We revealed that P, GF, and PFGF can alleviate alcoholic liver damage in chronically alcohol-fed mice. In addition, PFGF produced the best alleviation effect. PFGF could restore the intestinal barrier function by attenuating intestinal inflammation, thus reducing the amount of harmful substances entering the liver through the gut–liver axis, alleviating liver damage, and improving the liver’s function. Moreover, PFGF could restore the homeostasis of intestinal flora and increase the relative abundance of beneficial bacteria such as *Lactobacillus*, *Ruminococcaceae*, *Parabacteroids*, *Parasutterella*, and *Alistipes*, etc., to attenuate intestinal inflammation. However, the molecular mechanism of PFGF in improving alcoholic liver disease and the interaction of intestinal flora on alcoholic liver disease need further study and verification. Overall, this study provides a scientific basis for the processing and functional improvement caused by *Grifola frondosa*.

## Figures and Tables

**Figure 1 ijms-24-01406-f001:**
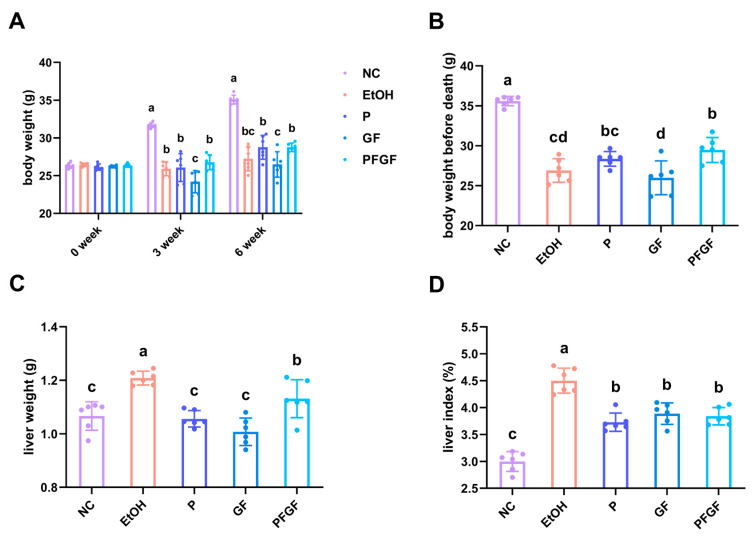
Effects of P, GF, and PFGF on body weight (**A**), body weight before death (**B**), liver weight (**C**), and liver index (**D**) in alcohol-fed mice. Data are expressed as mean ± SD (*n* = 6). Different superscript letters indicate statistically significant differences between groups (*p* < 0.05).

**Figure 2 ijms-24-01406-f002:**
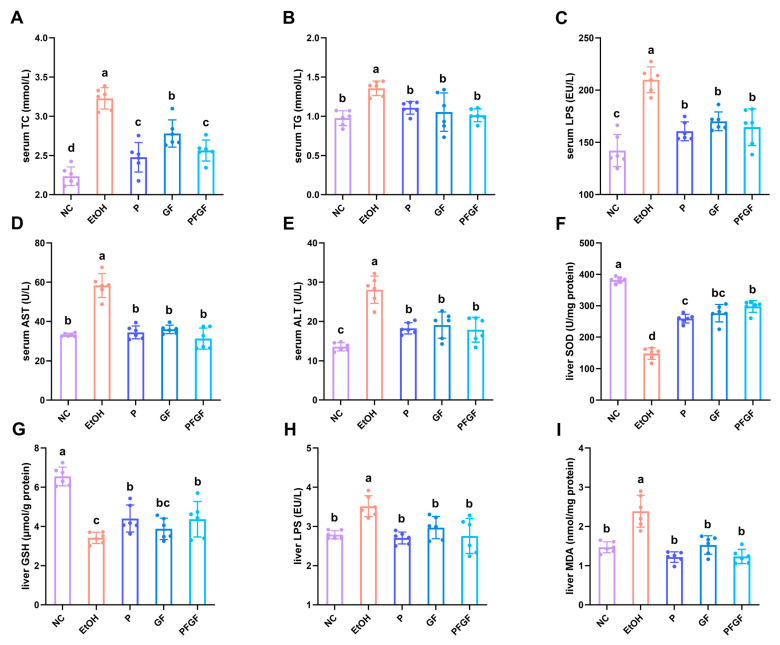
Effects of P, GF, and PFGF on serum TC (**A**), TG (**B**), LPS (**C**), AST (**D**), and ALT (**E**), and liver SOD (**F**), GSH (**G**), LPS (**H**), and MDA (**I**) in alcohol-fed mice. Data are expressed as mean ± SD (*n* = 6). Different superscript letters indicate statistically significant differences between groups (*p* < 0.05).

**Figure 3 ijms-24-01406-f003:**
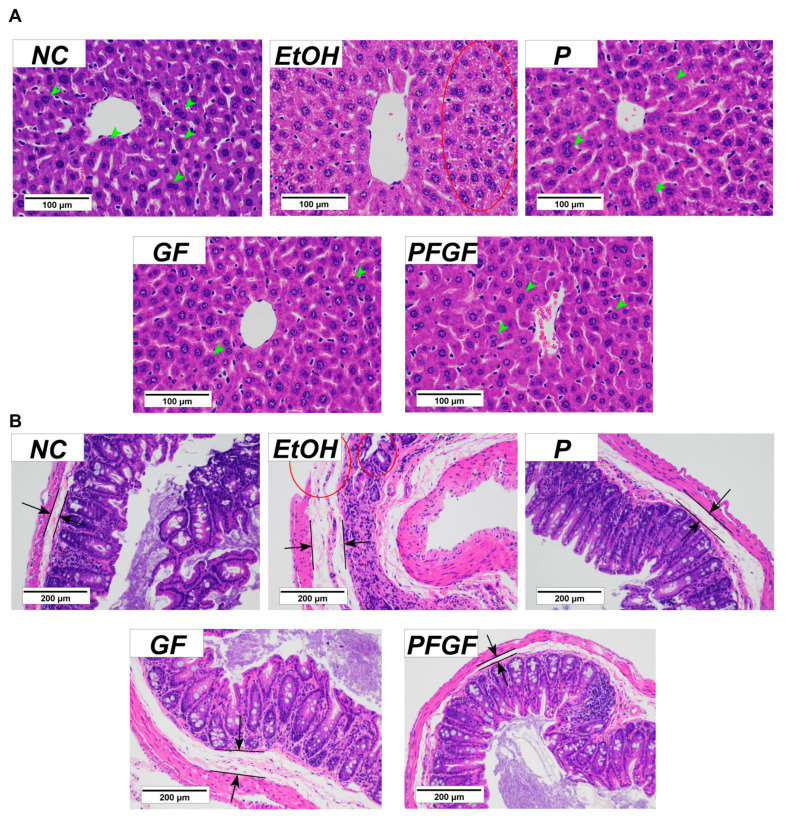
Results of histopathological analysis of liver (**A**) and cecal (**B**) tissues in different groups with hematoxylin and eosin staining at 400× and 200× magnification, respectively. Histopathologic characteristics are marked in the figure. Binucleate cells are marked by green arrows, lipid layer is marked by black arrows and tissue damage is marked by red circle.

**Figure 4 ijms-24-01406-f004:**
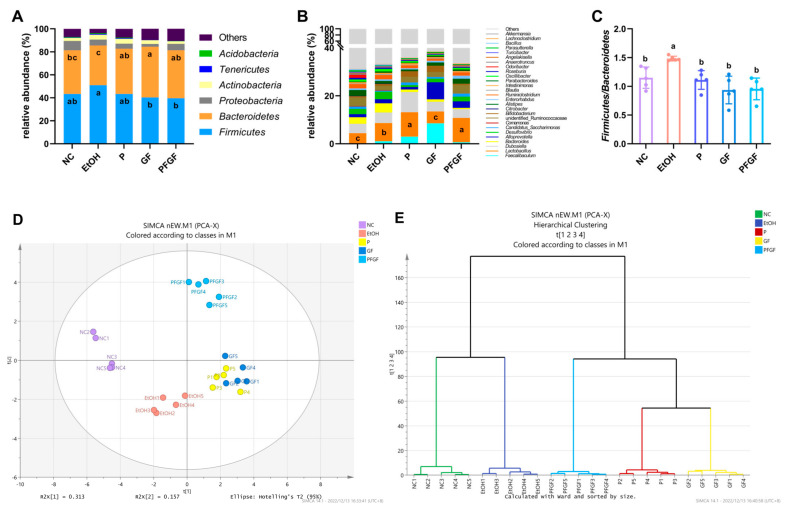
Components of intestinal microbiota at phylum (**A**) and genus (**B**) levels; value of relative abundance of *Firmicutes*/*Bacteroides* (**C**); PCA score (**D**); hierarchical clustering; (**E**) plots of intestinal microbiota at genus level in different groups. Different superscript letters indicate statistically significant differences between groups (*p* < 0.05).

**Figure 5 ijms-24-01406-f005:**
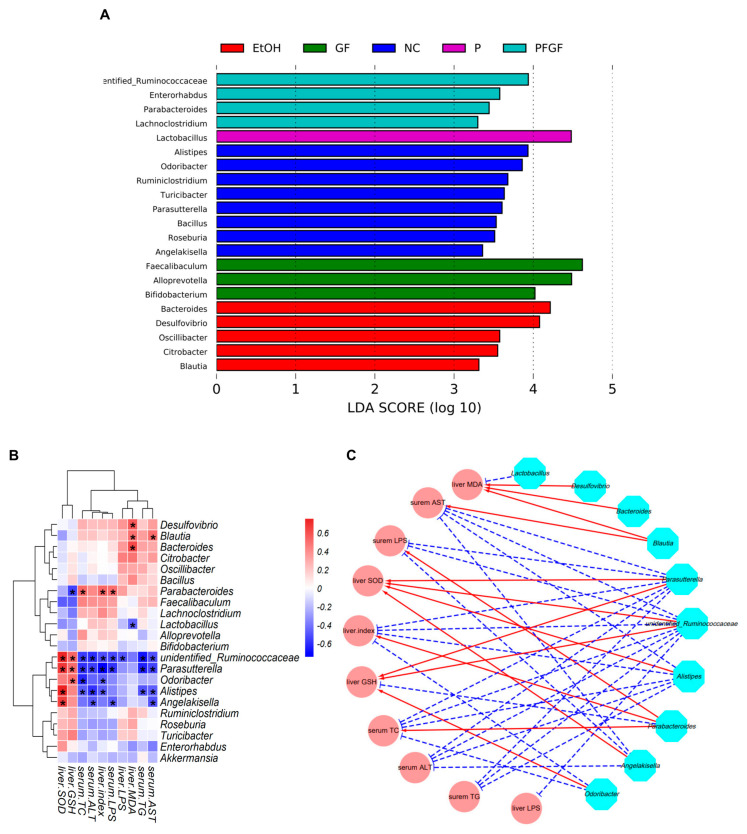
LEfSe analysis of intestinal microbiota at genus-level among NC, EtOH, P, GF, and PFGF groups (**A**). Heatmap of correlations between significant different intestinal microbiota and biochemical parameters (**B**); * indicates Spearman’s rank correlation coefficient > 0.5 or < −0.5. Visualization of correlation network. Cyan nodes: intestinal microbial genera; pink nodes: parameters; red lines: Spearman’s rank correlation coefficient > 0.5, adjusted *p* < 0.01; blue lines: Spearman’s rank correlation coefficient < −0.5, adjusted *p* < 0.01 (**C**).

**Table 1 ijms-24-01406-t001:** Composition and antioxidant activities of GF and PFGF; * indicates statistically significant differences (*p* < 0.05).

	TS mg/mL	RS mg/mL	PR μg/mL	TF mg/mL	TP mg/mL	LC mmol/L	DPPH %	FCA %	RPA OD Value
GF	86.29 ± 2.75	21.62 ± 0.16	310.46 ± 6.43	2.53 ± 0.36	14.09 ± 1.14	2.77 ± 0.05	77.22 ± 0.43	67.28 ± 0.29	0.586 ± 0.012
PFGF	56.21 ± 3.36 *	9.07 ± 0.09 *	286.88 ± 3.68 *	2.78 ± 0.45	15.15 ± 1.61	47.20 ± 0.29 *	80.98 ± 0.33 *	75.35 ± 0.31 *	0.622 ± 0.005 *

## Data Availability

Not applicable.

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
