# Peer review of "Probiotics-Fermented Grifola frondosa Total Active Components: Better Antioxidation and Microflora Regulation for Alleviating Alcoholic Liver Damage in Mice"

_ijms, 2023, doi:10.3390/ijms24021406_

Round 1

Reviewer 1 Report

Dear Authors:

Our main suggestions are the following:

1. The submitted manuscript is interesting, novel, and original. However, the English must be checked, the citations must be checked, and the manuscript must be checked for plagiarism with software to identify similar citations and be able to modify them. In addition, it is necessary to describe the introduction from the most general to the most specific (particular), always taking care to maintain coherence between the different paragraphs. Plagiarism has been detected.

2. Although abbreviations can be used, it must be ensured that they do not coincide with commonly known abbreviations. For example, FGF can also refer to a fibroblast growth factor.

3. Sentences can not start with an abbreviation.

4. The quality of figures 1–5 needs to be improved. The legibility of the texts included in the figures must also be ensured, specifically in figs. 3–5. Adequate scale bars in Fig. 3 are needed.

5. In "Figure 5", it needs to be bold.

6. Abbreviations must be defined upon their first appearance.

7. To improve the discussion and explain the results, it would be best to compare the antioxidant effects of the total active components of probiotic-fermented Grifola frondosa with those of other antioxidants, such as carotenoids like beta-carotene.

8. Is Table S1 for review such as non-published or supplementary material?

Waiting for comments and replies.

Kind regards,

Author Response

Responses for manuscript (ijms-2082052)

Dear Editor and Reviewers,

We greatly appreciate the time and effort you have devoted to reviewing our manuscript (Manuscript Number: ijms-2082052). To improve the readability, it is very important to explain our research in the best possible way. In this regard, your comments were quite thoughtful and very helpful. Thus, we have revised the manuscript based on your suggestions.

In the revised manuscript, a few paragraphs in certain sections have been modified to better explain the results and address the questions raised by the reviewers.

We have addressed the comments raised by the reviewers, and the revisions are marked up using the “Track Changes” function in the revised manuscript. Also, point-by-point responses to the reviewers’ comments are listed below this letter.

We hope that the revised version of the manuscript is now acceptable for publication in your journal.

Thank you very much for your patience and kindness. We look forward to hearing from you soon.

Yours sincerely,

Bin Liu

2022-12-18

Reviewer#1

Comment 1: The submitted manuscript is interesting, novel, and original. However, the English must be checked, the citations must be checked, and the manuscript must be checked for plagiarism with software to identify similar citations and be able to modify them. In addition, it is necessary to describe the introduction from the most general to the most specific (particular), always taking care to maintain coherence between the different paragraphs. Plagiarism has been detected.

Response: Thanks for your professional comment. Based on your suggestion, we have proofread and revised the manuscript. We have invited native English speakers with appropriate research background to make careful modifications too. We used iTheme/CrossCheck system to check for plagiarism, ensuring that the repetition rate of manuscripts does not exceed 30% and the single citation source does not exceed 6%. Language editing certificate and originality report are attached below.

Language editing certificate

originality report

Comment 2: Although abbreviations can be used, it must be ensured that they do not coincide with commonly known abbreviations. For example, FGF can also refer to a fibroblast growth factor.

Response: Thanks for your professional comment. We have changed the abbreviation of probiotic fermentation product of Grifola frondosa total active components to PFGF. It does not conflict with other commonly known publications in the biological or food technology field.

Comment 3: Sentences can not start with an abbreviation.

Response:Thanks for your professional comment. We have proofread and reviewed the manuscript to ensure that this problem does not appear in the manuscript.

Comment 4:The quality of figures 1–5 needs to be improved. The legibility of the texts included in the figures must also be ensured, specifically in figs. 3–5. Adequate scale bars in Fig. 3 are needed.

Response:Thanks for your professional comment. We increased the image quality and re typesetting to make the image and text clearer.

Comment 5:In "Figure 5", it needs to be bold.

Response:Thanks for your professional comment. We have proofread and reviewed the manuscript to ensure that this problem does not appear in the manuscript.

Comment 6:Abbreviations must be defined upon their first appearance.

Response:Thanks for your care and patience. We have proofread and reviewed the manuscript to ensure abbreviations be defined upon their first appearance.

Comment 7:To improve the discussion and explain the results, it would be best to compare the antioxidant effects of the total active components of probiotic-fermented Grifola frondosa with those of other antioxidants, such as carotenoids like beta-carotene.

Response:Thanks for your professional comment. In the pre experiment, we compared the antioxidant activity of various extracts of GF with that of ascorbic acid, and found that they had good antioxidant activity (Zhu, 2022, dissertation, unpublished). In this study, we mainly compared the antioxidant activity changes of GF and FGF, the antioxidant activity of FGF was significantly improved compared with GF. We didn't set up a positive drug control, but we will use it in future experiments.

Comment 8:Is Table S1 for review such as non-published or supplementary material?

Response:Thank you for your valuable and thoughtful comments. Table S1 is also applicable to review. All data that support the findings of this study are non-published and available from the corresponding author upon reasonable request.

Reviewer 2 Report

It is suitable for the scope of the International Journal of Molecular Scineces. However, the following points need to be revised to enhance the reader’s understanding. 

1. Please describe the IACUC number of animal experiments
.

2. Please provide high resolution figure 5A, 5B, 5C.

3. As the authors mentioned in the conclusion section, please provide own research results or references related to FGF inhibits intestinal inflammation.

Author Response

Responses for manuscript (ijms-2082052)

Dear Editor and Reviewers,

We greatly appreciate the time and effort you have devoted to reviewing our manuscript (Manuscript Number: ijms-2082052). To improve the readability, it is very important to explain our research in the best possible way. In this regard, your comments were quite thoughtful and very helpful. Thus, we have revised the manuscript based on your suggestions.

In the revised manuscript, a few paragraphs in certain sections have been modified to better explain the results and address the questions raised by the reviewers.

We have addressed the comments raised by the reviewers, and the revisions are marked up using the “Track Changes” function in the revised manuscript. Also, point-by-point responses to the reviewers’ comments are listed below this letter.

We hope that the revised version of the manuscript is now acceptable for publication in your journal.

Thank you very much for your patience and kindness. We look forward to hearing from you soon.

Yours sincerely,

Bin Liu

2022-12-18

Reviewer#2

Comment 1: Please describe the IACUC number of animal experiments.

Response: Thanks for your professional comment. This research was performed in accordance with the Guidelines for Care and Use of Laboratory Animals of Fujian Agriculture and Forestry University and supervised by the animal experimental ethics committee. (Approval number: JC-2021-020).

The scanned copy of the table of animal experimental ethical inspection:

Comment 2: Please provide high resolution figure 5A, 5B, 5C.

Response: Thanks for your professional comment. We improved the definition of figure 5A, 5B and 5C and adjusted the typesetting. Now the figures and texts are clearer.

Comment 3: As the authors mentioned in the conclusion section, please provide own research results or references related to FGF inhibits intestinal inflammation.

Response:Thanks for your professional comment. In our previous study, the compounds added with Grifola frondosa total active components (GF) improved the jejunum tissue and reduced the level of IL-6 in of type 2 diabetes mice, which indicates that GF can alleviate intestinal inflammation (He et al. doi.org/10.1016/j.algal.2022.102791). In this stude, with the intraperitoneal administration of FGF as was the integrity of the intestinal wall. A thin lipid layer was maintained in FGF, but the GF group showed thickening of the lipid layer. These show that GF can alleviate intestinal inflammation, and FGF has a better effect.

Original sentences:

The cecal tissue of the NC group was clearly stained. The cecal villi were intact and neatly arranged and cecal wall was intact with a thin lipid layer. The cecal tissue of the EtOH group showed significant inflammation. The cecal villi were broken, the cecal wall was damaged, and the lipid layer was thickened. With intraperitoneal administration, the P, GF and FGF groups maintained the integrity and orderliness of the intestinal villi and the integrity of the intestinal wall. Both the P and FGF groups maintained a thin lipid layer but the GF group still showed thickening of the lipid layer.

Histopathology of cecal tissue can reflect the states of the intestinal barrier and inflammation in intestine. The results of cecal tissue sections showed that the P, GF and FGF had protective effects on the intestinal barrier.

Revised sentences:

The cecal tissue of the NC group was clearly stained. The cecal villi were intact and neatly arranged, and the cecal wall was intact, with a thin lipid layer. The cecal tissue of the EtOH group showed substantial inflammation. The cecal villi were broken, the cecal wall was damaged, and the lipid layer was thickened. With intraperitoneal administration, the integrity and orderliness of the intestinal villi in the P, GF and PFGF groups were maintained, as was the integrity of the intestinal wall. A thin lipid layer was maintained in both the P and PFGF groups, but the GF group showed thickening of the lipid layer.

The histopathology of cecal tissue can reflect the states of the intestinal barrier and inflammation in the intestine. In our previous study, the compounds added with GF improved the jejunum tissue and reduced the level of IL-6 in of type 2 diabetes mice, which indicates that GF can alleviate intestinal inflammation [45]. The results of the cecal tissue sections showed that the P, GF and PFGF treatments had protective effects on the intestinal barrier.

Round 2

Reviewer 1 Report

Dear Authors:

The manuscript is now appropriate for publication. However, the following changes must be made before:

1. Italics must be used where appropriate, specifically in figs. 4 and 5. 

2. In the legend of "Figure 5", it needs to be bold.

3. Why the results of Zhu, 2022 have been not published?

Kind regards,

Author Response

Comment 1: Italics must be used where appropriate, specifically in figs. 4 and 5. 

Response: Thanks for your professional comment. We have marked the flora names in the figure in italics except Fig. 5A. This is due to the fact that Fig. 5A is exported by Huttenhower Lab Galaxy server(http://huttenhower.sph.harvard.edu/lefse/). We can't change the font of Fig. 5A.

Comment 2: In the legend of "Figure 5", it needs to be bold.

Response: Thanks for your care and patience. We have bold "Figure 5".

Comment 3: Why the results of Zhu, 2022 have been not published?

Response:Thanks for your professional comment. The result of Zhu, as a dissertation, has been uploaded to CNKI, but is not online. Again, we're sorry we didn't have to set up a positive drug control, and we will set it in future experiments.